# Dynamic Changes in Ecosystem Service Value and Ecological Compensation in Original Continuous Poverty-Stricken Areas of China

**Zhe Yu [1,2], Chunwei Song [3,4,*] and Huishi Du [3,*]**

[1] School of Chinese Language and Literature, Jilin Normal University, Siping 136000, China; 13204492670@163.com
[2] Ministry of Human Resources and Social Security of the People's Republic of Tiedong District of Siping City, Siping 136000, China
[3] College of Geographical Sciences and Tourism, Jilin Normal University, Siping 136000, China
[4] College of Forestry, Northeast Forestry University, Harbin 150040, China
[*] Correspondence: schw1817@163.com (C.S.); duhs@jlnu.edu.cn (H.D.)

**Abstract:** The original contiguous poverty-stricken areas of China (OCPSAC) are a regional complex with natural and human attributes. Their ecosystem services are highly complex in their contribution to human well-being. The region's ecosystem is unstable and socio-economic development is unbalanced, making it a key area for research and the implementation of the United Nations Sustainable Development Goals. Therefore, it has become the focus of social attention. This study obtained satellite remote sensing images of the Landsat8 based on the Google Earth Engine. The dynamic characteristics of the spatial and temporal distribution of the OCPSAC's land use changes in 2015 and 2022 were analyzed. The characteristics of the changes in ecosystem service value (ESV) were quantitatively evaluated through the revised ESV equivalent scale, and the ecological compensation (EC) standards and EC priority level suitable for the region were explored. The research results showed that: (1) the OCPSAC land use types were mainly grassland, forests, and deserts, accounting for more than 80% of the research area; (2) the OCPSAC's ESV reached CNY 115.578 billion, and the Tibet Region (TR) and Tibetan Region of Four Provinces (TRFP) had the highest ESVs, accounting for 32.28% and 19.64%, respectively. Among individual ESVs, hydrological regulation (HR) and climate regulation (CR) had the highest values, accounting for 55.23% of the research area; (3) the ESV of the research area presented the characteristics of being high in the middle and low at both ends. From the terrain perspective, the areas with a higher ESV per unit area were distributed in the second altitude gradient, followed by China's third altitude gradient, and the first altitude gradient was the lowest; (4) in the period of 2015–2022, the ESV increased by CNY 21.39 billion, of which TR's ESV increased by CNY 17.44 billion. From the perspective of a single ESV, the value of waste treatment (WT), HR, nutrient cycle (NC), and provide aesthetic landscape (PAL) increased by CNY 24.38 billion; and (5) in 2022, the EC standard of the OCPSAC was CNY 917.14 billion. High-EC-value areas were mainly distributed in TR and TRFP, accounting for 72.79% of the total compensation amount, of which only TR's EC amount exceeded a GDP of 2022, accounting for 4.77% of the total compensation. The purpose of this study was to provide a reference for ecosystem sustainability and EC.

**Keywords:** ecosystem service value (ESV); ecological compensation (EC); sustainable development; rural revitalization; original continuous poverty-stricken areas of China (OCPSAC)

## 1. Introduction

In September 2015, the United Nations 2030 Agenda for Sustainable Development proposed 17 Sustainable Development Goals (SDGs), with SDG 15 (life on land) as the core. It proposed sustainable development policies covering three aspects: mountain economy, society, and environment. As human activities in mountainous areas are more limited by

changes in altitude, slope, precipitation, and temperature, these areas have become a key area for research and the implementation of the United Nations SDGs [1–3]. The original contiguous poverty-stricken areas of China (OCPSAC) are mainly deep mountain and rocky areas, high and cold areas, ecologically fragile areas, disaster-prone areas, and ecological protection areas [4–6]. The ecosystem of the OCPSAC has obvious independence and vulnerability, and when the ecological environment is destroyed, it will have irreversible consequences [7–9]. The OCPSAC covers most poor areas and deep poverty groups. By implementing targeted poverty alleviation policies, according to China's current standards, rural low-income households will achieve poverty alleviation, all poverty-stricken counties and regional overall poverty will be resolved, and the poverty reduction goal of the United Nations 2030 Agenda for Sustainable Development will be achieved 10 years earlier. However, the natural conditions of the OCPSAC are poor, the infrastructure is weak, industrial development lags, it is difficult for farmers to increase their income, and there is a risk of a large-scale return to poverty [10–12]. Judging by existing OCPSAC research results, scholars mainly focus on research on the mechanisms [13], causes [14], and policies [15] of rural poverty. However, there is relatively little research on ecological compensation (EC) standards for rural poverty alleviation and it is not combined with sustainable development and rural revitalization strategies. Therefore, scientifically and accurately assessing the value of regional ecological services and formulating regional EC development standards will not only comprehensively measure regional benefits and promote regional sustainable development, but also play a vital role in consolidating and expanding the organic link between poverty alleviation and rural revitalization and making follow-up decisions on rural revitalization [16–19].

Ecosystem services value (ESV) refers to the estimation of ecosystem services and the economic laws of natural capital. It mainly manifests itself in the environment formed by the land system and is used to maintain human survival and development, and directly or indirectly provides production, daily necessities, and services, including maintaining atmospheric balance, climate regulation, the provision of food and water, and maintaining biodiversity, which is the value of the various services obtained by human beings from the natural environment [20–22]. In recent years, scholars have estimated and predicted ESVs at different spatial scales, such as countries [23], regions [24], and river basins [25,26] based on different research methods. Arowolo et al. [27] used the value transfer method to analyze the changes in different land use types in Nigeria in 2000 and 2010 to explore the ESV spatial patterns and temporal changes. Aziz [28] used the value transfer method to analyze the spatial and temporal evolution characteristics of Pakistan's overall ESV from the perspective of land use types. Qu et al. [29] used a land use spatial analysis model and meta-analysis method to study the ecological land and changing spatial pattern of Jinggang Mountain in the midstream zone of Luoxiao Mountain (LXM), and the profit and loss of the ESV. In summary, land use research has the advantages of easy data acquisition, simple operation, and strong applicability for ESV assessment, but its accuracy in practical application is not high, and other methods need to be introduced to correct it. This study corrects ecological indicators from Net Primary Productivity (NPP) and precipitation and soil conservation (SC) according to the equivalent factor method in Xie et al.'s [30] study.

Ecological compensation (EC) studies investigate costs, values, and benefits, aim to protect the environment, promote human and natural development, and provide ecological payment to ecosystem services and ecological product suppliers through economic means [31,32]. At present, the research on EC focuses on the theoretical basis [33], institutional construction [34], and compensation model [35], focusing on the survey of ecological willingness to pay. Rasheed et al. [36] estimated the ESV of rice based on market price, replacement cost, and benefit transfer methods, and used a single-boundary binary method to estimate the EC potential in protecting agricultural biodiversity. Shang et al. [37] investigated the ESV of the El Lake Basin in 2000–2015 based on the InVEST model and adjusted the EC policy according to different land use standards. Based on the EC model of Payments for ecosystem services (PES), Niu et al. [38] determined the subject and object of

different ecosystem compensations according to the spatial and temporal relationship between different ecosystem adjustment services and economic systems in Fuzhou, and then calculated the EC quota of Fuzhou in the period of 2015–2018. In summary, EC is mostly concentrated on regional or administrative regional studies. The measurement standards of EC are often calculated according to the current value of ecosystem services. Because the equivalent coefficient of ESV has not been adjusted, it is impossible to accurately reflect the spatial heterogeneity in regional or administrative areas. However, the EC method is affected by subjective factors. In order not to be affected by this, it accurately reflects the non-market value of ecosystem services based on land use data when calculating the ESV to ensure that the regional EC quota is the best, which can effectively help stakeholders to formulate appropriate ecological compensation policies.

The OCPSAC is an important water conservation area and ecological barrier in China. It is rich in mineral resources, and its unique geospatial location makes its habitat diverse [39]. Due to the impact of economic and social development and human activities, leaky mining and tailings reservoirs have completely destroyed the natural landscape of the OPCSAC, resulting in a series of environmental problems such as soil acidification, air pollution, groundwater level decline, and biodiversity reduction, which seriously affect the ecological balance. Many of the 14 continuous special hardship areas are ecological security strategic areas built by the state. For example, some areas in the national "two screens and three belts" are the "Ecological barrier of Qinghai-Tibet Plateau", "Ecological Barrier of the Loess Plateau Sichuan and Yunnan Ecological Barrier", and "Southern Hilly and Mountain Zone" [40,41]. Therefore, the importance of the ecological function of the OCPSAC is to promote the optimization of the compensation strategy of the OCPSAC, improve the efficiency and fairness of its EC, and accelerate the harmonious coexistence between man and nature through scientific analysis of its ESV, EC priority, and EC amount. In addition, EC, as an effective means to solve human–land conflicts in ecologically fragile areas, plays an important role in promoting the economic and social stability and sustainable development of ecologically fragile areas. This study takes the OCPSAC as the research area to correct the functional value of each individual ecosystem with the three aspects of NPP, precipitation, and SC, and then calculates the spatial and temporal evolutionary characteristics of the ESV. Based on the conversion coefficient method, the regional compensation intensity coefficient is enhanced, the EC standard is calculated through the non-market amount, and the EC amount differentiation is carried out according to the ESV characteristics of the research area and local conditions. This research aims to provide a scientific reference for sustainable development and rural revitalization strategies in China.

## 2. Study Area

The OCPSAC refers to the continuous division of poverty-stricken counties with natural geography, similar climate and environmental conditions, traditional industries, cultural customs, and poverty-causing factors, mainly including Liupanshan Mountain (LPSM), Qinling-Dabashan Mountain (QDM), Wuling Mountain (WLM), Wumeng Mountain (WMM), Yunnan-Guizhou-Guangxi Rock Desert Mountain (YGGM), Western Yunnan Border (WYB), South Daxinganling Mountain (SDM), Yanshan-Taihangshan Mountain (YTM), Lyuliang Mountain (LLM), Dabie Mountain (DBM), Luoxiao Mountain (LXM), Tibet Region (TR), Tibetan Region of Four Provinces (TRFP), and Three prefectures of Xinjiang (TPXJ) (Figure 1). The OCPSAC covers 680 counties in 22 provinces and cities in China, with a total land area of 4.02 million km$^2$, representing 42.52% of the country's total land area. The landforms in the study area are mostly plateaus, mountains, hills, and plains, and the climate types are mainly temperate and subtropical monsoon climates, with an average forest coverage rate of 38.90%. Most areas are rich in water resources, SDM, YTM, YGGM, WLM, QDM, LLM, and WMM are relatively rich in mineral resources, and YGGM, WLM, and QDM are rich in biological species and tourism resources. The OCPSAC is mainly a marginal area of China's inter-provincial border, and most of it is located between transition zones and junction areas of different landform types. It is under

the jurisdiction of many provinces and regions, making it a watershed in the regional economy. The OCPSAC has problems such as weak public infrastructure, a low-quality workforce, an insufficient supply of public services, regional self-development capacity, a serious lack of capacity for sustainable development, and a backward level of economic and social development [42,43].

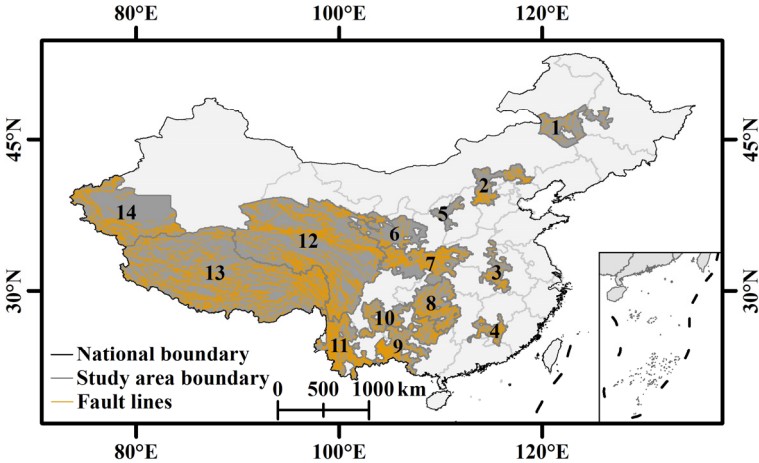

**Figure 1.** Spatial distribution of the OCPSAC (1. SDM; 2. YTM; 3. DBM; 4. LXM; 5. LLM; 6. LPSM; 7. QBM; 8. WLM; 9. YGGM; 10. WMM; 11. WYB; 12. TRFP; 13. TR; and 14. TPXJ).

## 3. Materials and Methods

### 3.1. Data Sources and Processing

The period of poverty alleviation began in 2013. In November 2015, the Central Committee of the Communist Party of China and the State Council issued the Decision on Winning the Battle of Poverty Alleviation, which officially took targeted poverty alleviation as the basic strategy of poverty alleviation and development and began to fully start this major battle of great historical significance. On 25 February 2021, China won an all-round victory in the battle against poverty. In March, the Opinions of the Central Committee of the Communist Party of China and the State Council on Realizing the Effective Connection between Consolidating and Expanding the Achievements of Poverty Alleviation and Rural Revitalization was publicly released, which clearly pointed out that, after the task of poverty alleviation is completed, a five-year transition period will be established. Its main task is to consolidate the effectiveness of poverty alleviation, prevent poverty return, and explore a long-term mechanism to solve relative poverty. In 2022, through the joint efforts of all parties, the achievements of poverty alleviation were further consolidated and expanded, and the bottom line of large-scale poverty alleviation was maintained. New progress has been made in rural development, rural construction, and rural governance. Therefore, we chose 2015 and 2022 as the research period.

The land use data in this study were obtained from the GEE Cloud platform to acquire Landsat 8 image data for 2015 and 2022 covering the study area, and the image space resolution was 30 m. API programming was used for image preprocessing, such as radiation correction, image splicing, and atmospheric apparent reflectivity, etc. [44,45]. Six bands such as 2, 3, 4, 5, 6, and 7 of the Landsat 8 satellite were used for image classification calculation. This study is based on the national standard Land Use Status Classification organized and revised by the Ministry of Natural Resources as land use classification [29]. Combined with the characteristics of land use in the OCPSAC, land use was classified, where the first level was divided into 6 categories and the second level was divided into 23 categories (Table 1). According to the research needs, ArcGIS 10.8 was used to analyze the six types of land used in the primary classification, such as farmland, forest, grassland, wetland, sandy, and others, in order to observe the changes in the land use structure of the original concentrated and continuous special hardship areas in China. To ensure the accuracy of the data processing,

the training sample data and accuracy verification sample data mainly came from the visual identification of expert knowledge in Google Earth Pro software (v7.3.6.9796), because Google Earth Pro software (v7.3.6.9796) not only has satellite images, maps, terrain, and architecture, but also rich data layers such as objects and 3D views, and allows you to view past and current Earth images. In total, 70% of the sample points were used for classifier training and 30% for precision testing. It was verified that the overall accuracy rate exceeded 80% [46,47]. The DEM data came from the Resource and Environmental Science and Data Center of the Chinese Academy of Sciences (https://www.resdc.cn/ accessed on 19 September 2023) with a resolution of 90 m. Net Primary Productivity (NPP) data came from NASA's annual NPP data set (http://modis.gsfc.nasa.gov/ accessed on 19 September 2023), and the image space resolution was 30 m. Average annual rainfall data came from the China Meteorological Science Data Sharing Service Network (http://cdc.cma.gov.cn/ accessed on 19 September 2023), which were sorted to obtain the average annual rainfall of each site, and multi-year average precipitation data with a 30 m resolution were obtained through the spatial interpolation method for the site data. Soil attribute data and soil type distribution data came from the China Soil Database of Nanjing Soil Institute(1:1 million) (http://www.soil.csdb.cn/ accessed on 24 September 2023). Finally, data on GDP, grain area output, and agricultural product price came from the China Statistical Yearbook available on the website of the National Bureau of Statistics of the People's Republic of China (http://www.stats.gov.cn/ accessed on 10 October 2024). Among them, the GDP data of the OCPSAC came from the statistics of counties, and the grain area output and agricultural product price were calculated based on national statistics. To reduce errors caused by research data such as spatial resolution and coordinate systems, this paper uniformly projected all data to the WGS 84 coordinate system and used ArcGIS 10.8 software to resample the data resolution to 30 m through the nearest method for data processing and analysis.

**Table 1.** Classification of land use types.

| Primary Classification | Secondary Classification |
| --- | --- |
| Farmland | Dry land; paddy field |
| Forest | Woodland; shrubland; sparse woodland; other woodland |
| Grassland | High-vegetation-covered grassland; medium-vegetation-covered grassland; low-vegetation-covered grassland |
| Wetland | Rivers; lakes; reservoirs; beaches; permanent glaciers and snow |
| Desert | Sandy land; Gobi |
| Others | Urban land; rural settlements; other construction land; swamp land; saline and alkaline land; bare land; gravel land |

### 3.2. ESV Calculation

This study divides ecosystem services into 10 types: food production (FP), raw material production (RMP), gas regulation (GR), climate regulation (CR), waste treatment (WT), hydrological regulation (HR), soil conservation (SC), biodiversity protection (BP), nutrient cycle (NC), and provide aesthetic landscape (PAL) [48–50]. When using the equivalent factor method for evaluating the regional ESV, it is necessary to modify the equivalent factor according to different climates, biomasses, terrains, and other factors in the region [51]. Referring to the research of previous scholars [52], the ecosystem functions of FD, RMP, GR, CR, WT, BP, NC, and PAL are generally positively related to biomass, HR functions are related to precipitation changes, and SC functions are closely related to precipitation, terrain slope, soil properties, and vegetation coverage [53]. Based on this, in combination

with the ESV basic equivalent scale, a spatial and temporal dynamic change value scale of ecological services is constructed based on the following formula:

$$F_{nij} = P_{ij} \times F_{n1} + R_{ij} \times F_{n2} + S_{ij} \times F_{n3} \tag{1}$$

where $F_{nij}$ is the value equivalent factor per unit area of the ecosystem for the $n$-th type of ecosystem service function in the $j$-th area in year $i$; $P_{ij}$, $R_{ij}$, and $S_{ij}$ refer to the spatio-temporal regulators of NPP, precipitation, and SC in region j and year i of the ecosystem class, respectively; $F_n$ refers to the $n$-th ecosystem service value equivalent factor for that ecosystem type; and $n_1$, $n_2$, and $n_3$ refer to NPP-, precipitation-, and SC-related services, respectively.

The NPP spatio-temporal modifier is mainly used to correct the ecological service value equivalent factor for FD, RMP, GR, CR, WT, BP, NC, and PAL. The spatio-temporal adjustment factor for precipitation is primarily an ecological service value equivalent factor that is used to correct for HR service functions. The SC spatio-temporal adjustment factor is mainly an ecological service value equivalent factor applied to correct SC service functions [54–56]. The formula is as follows:

$$P_{ij} = \frac{B_{ij}}{B} \tag{2}$$

$$R_{ij} = \frac{W_{ij}}{W} \tag{3}$$

$$S_{ij} = \frac{E_{ij}}{E} \tag{4}$$

where $B_{ij}$, $W_{ij}$, and $E_{ij}$ refer to the NPP (t/ha) of the $j$-th region of the ecosystem class in year $i$, respectively; the mean amount of precipitation per unit area (mm/ha) and $E_{ij}$ refer to the SC simulation for area $j$ in year $i$ of the ecosystem type; and $B$, $W$, and $E$ denote the mean NPP (t/ha), mean annual rainfall per unit area (mm/ha), and mean SC simulation per unit area for this type of ecosystem in the study area.

The potential ability of different ecosystem types to contribute to ecosystem service functions varies considerably. The one standard unit ESV equivalent factor refers to the economic value of the annual natural grain output of 1 ha of farmland with the average national output. Based on the results of the classification [57], the ESV equivalent table with characteristics based on Chinese national conditions can be determined [58]. This study estimates that the economic value of a standard ESV equivalent factor in the research area is equivalent to the market value of 1/7 of the average grain yield in the country in that year. Due to the spatial difference in the ESV of the standard equivalent factor, to reduce the impact of inflation on the evaluation results, this paper calculates the average price of China's major grains and the average output of major grain crops in 2015 and 2022. Combined with the regional correction coefficient of ESV, the average grain yield in 2015 and 2022 was 5539 kg/ha and 5805 kg/ha, respectively, and the average unit price of grain was 2.64 CNY/kg and 3.21 CNY/kg, respectively. The economic value of one standard ESV equivalent factor in 2015 and 2022 was calculated to be about 2373.57 CNY/ha, and then the OCPSAC's ESV equivalent table was obtained (Table 2). The formula is as follows:

$$ESV_f = \sum_{i=1}^{n} \left( A_i \times VC_{fi} \right) \tag{5}$$

$$ESV = \sum_{i=1}^{n} A_i \times VC_i \tag{6}$$

where $ESV$ represents the total service value of the ecosystem in the research area; $A_i$ represents the area (ha) of the $i$ type land use in the research area; $VC_i$ represents the ESV coefficient of the $i$ type land use; $ESV_f$ represents the value of the item f service function of

the ecosystem; and $VC_{fi}$ represents the ESV coefficient of item *f* of the *i* land use type in the research area.

**Table 2.** The OCPSAC's ESV equivalent table (CNY/ha).

| Functional Value | Farmland | Forest | Grassland | Wetland | Desert |
|:---:|:---:|:---:|:---:|:---:|:---:|
| FD | 2634.66 | 593.39 | 545.52 | 1044.37 | 23.74 |
| RMP | 593.39 | 1376.67 | 807.01 | 569.66 | 71.21 |
| GR | 2112.48 | 4533.52 | 2872.02 | 2254.89 | 308.56 |
| CR | 1115.58 | 13,553.08 | 7571.69 | 5221.85 | 237.67 |
| WT | 332.30 | 3963.86 | 2492.25 | 7358.07 | 973.16 |
| HR | 3560.36 | 8877.15 | 5554.15 | 105,695.07 | 569.66 |
| SC | 1234.26 | 5506.68 | 3489.15 | 1946.33 | 2207.42 |
| BP | 379.77 | 427.24 | 261.09 | 189.89 | 23.74 |
| NC | 403.51 | 5031.97 | 3251.79 | 8260.02 | 332.30 |
| PAL | 189.89 | 2207.42 | 1400.41 | 5316.80 | 142.41 |

*3.3. Calculation of the EC Standard Quota*

The GDP growth of the OCPSAC mainly derives from the development and utilization of natural resources, and the lower the GDP, the more urgent the need for economic development in the district [59]. The EC priority can determine the urgency of ecological compensation. Since FP and RMP are monetized to contribute to the economic development of the region, the non-market economic value of GR, CR, HR, WT, SC, NC, BP, and PAL is selected for the calculation of the ecological compensation priority. The formula is as follows:

$$P = \frac{V_n}{G_n} \tag{7}$$

where *P* stands for the EC priority; $V_n$ stands for the regional ESV non-market value; and $G_n$ stands for the regional GDP. P $\geq$ 1 is the EC priority area, and the higher the value of P, the higher the amount of EC that should be compensated. When P < 1 is the non-EC priority area, the smaller the value of P, the lower the amount of EC.

EC standards should consider factors such as the level of national economic development, the importance of ecological location, and compensation for regional ecological quality [60]. Based on the conversion coefficient method, this study selects 15% as the ecological value conversion coefficient of the OCPSAC with reference to the existing cases [61]. This can not only reduce the urgent development needs of economically backward areas, but also reduce the risk of a rapid reduction in ESV in the region. To avoid the EC being too centralized, a tangent function was introduced to normalize the EC priorities and use ecosystem compensation coefficients to reflect regional differences in EC standards. The formula is as follows:

$$K = \frac{2\arctan P}{\pi} \tag{8}$$

$$T = V_n \times R \times K \tag{9}$$

where *K* denotes the regional EC intensity factor; *T* is the regional EC standard; *P* is the EC priority; and *R* is the ecological value conversion coefficient of 15%.

*3.4. ESV Change Calculation*

The change in the ESV is found by comparing the differences in the values of ecosystem services in different periods [20]. The calculation formula is as follows:

$$ESV_c = \frac{ESV_j - ESV_i}{ESV_i} \times 100\% \tag{10}$$

where $ESV_c$ is the change range of the ESV and $ESV_i$ and $ESV_j$ are the ecosystem service values in the early and final stages of the research period, respectively.

## 4. Results

### *4.1. Spatial and Temporal Changes in the OCPSAC Land Use Type*

The OCPSAC land use types had similar spatial distributions in 2015 and 2022 (Figure 2a,b). In 2022, the OCPSAC was dominated by grassland, forest, and desert areas, which together accounted for 84.28% of the study area, with grassland accounting for the largest area with approximately 155 ha. Farmland was mainly distributed in SDM, LPSM, and QDM, accounting for about 37.44% of the farmland area in the research area. Forest was mainly distributed in TR, TRFP, YGGM, WYB, and WLM in the south, accounting for about 71.46% of the total forest area in the research area. Grassland was mainly distributed in TR and TRFP, accounting for more than 70% of the total grassland area in the research area. Wetland and desert were mainly located in the western part of the study area in TRFP, TR, and TPXJ, and accounted for 88.59% and 97.97% of the total wetland and desert area in the research area, respectively. Other lands were mainly distributed in TRFP and DBM, which together accounted for 65.51% of the other land area in the study area. In 2015 and 2022, the OCPSAC land use types showed different dynamics, with wetland and other land showing an expanding trend, while farmland, forest, grassland, and desert areas showed varying degrees of contraction. Regarding the change in land use types, the areas of wetland and other land increased by 1.42% and 2.86%, respectively; the wetland area in YGGM, WMM, and WYB increased by 21.70%, 17.10%, and 12.62%, respectively, while the wetland area in SDM decreased by 10.24%. The areas of other land in LXM, TR, and LLM increased by 22.52%, 23.33%, and 16.74%, respectively. The areas of farmland, forest, grassland, and desert decreased by less than 1%, of which the desert area in LXM changed significantly, with a decrease of 32.94%, while the changes in other areas of the OCPSAC were smaller, with a decrease of 0.20%. Among them, farmland was mainly transformed into forest and grassland, and other land was mainly transformed into farmland and grassland. The area of farmland and other land transformed into grassland was about 6.97% (Figure 2c).

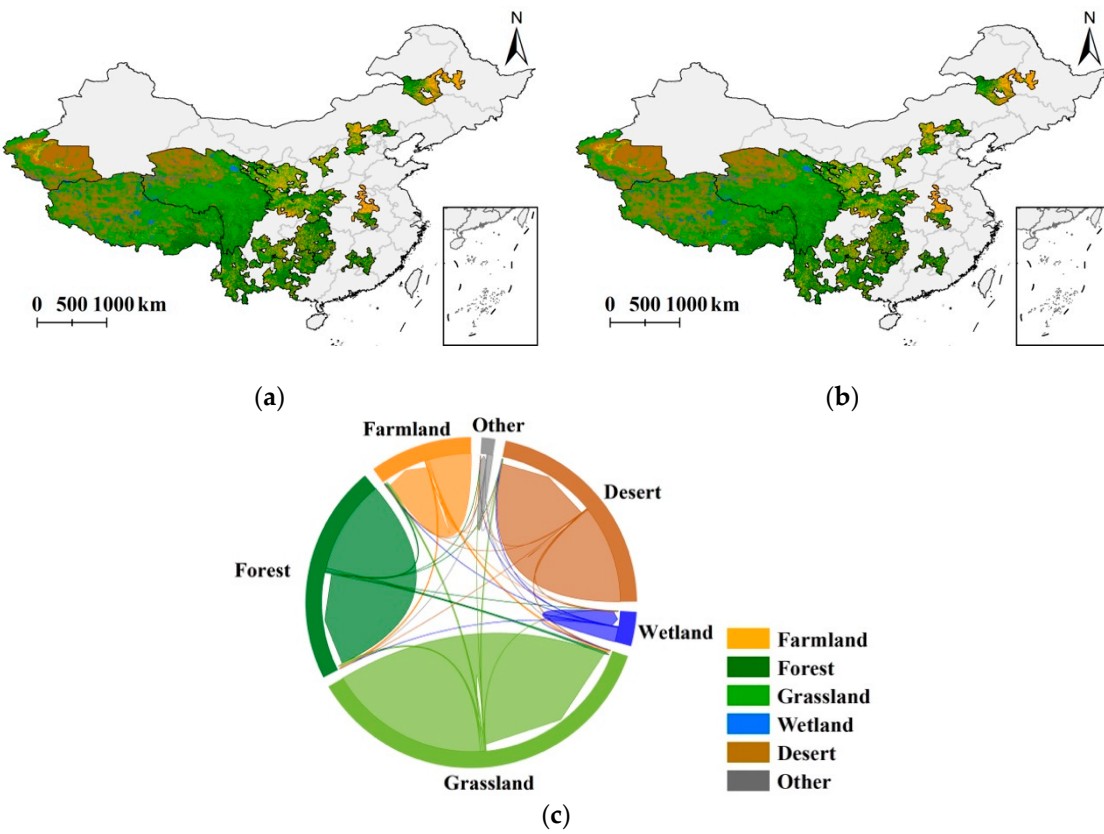

**Figure 2.** Spatial distribution of land use types ((**a**) 2015; (**b**) 2022; and (**c**) 2015–2022).

### 4.2. ESV Status of the OCPSAC

In 2022, the total ESV in the OCPSAC reached CNY 11,559.78 billion (Figure 3a). The ESVs of TR and TRFP were higher, at more than CNY 1000 billion, of which TR's ESV was the highest, reaching CNY 3614.90 billion, accounting for 30.28% of the total ESV in the research area, followed by TRFP, whose ESV was CNY 2770.51 billion. The ESVs in the five regions of WYB, YGGM, WLM, QDM, and TPXJ were between CNY 500 and 1000 billion. The ESVs in the seven areas of WMM, LSPM, SDM, LXM, DBM, LLM, and YTM did not exceed CNY 500 billion, of which LLM's ESV was the lowest, at only CNY 91.26 billion, accounting for about 0.76% of the total ESV. The functional value of different ecosystem services of the OCPSAC varied significantly (Figure 3b). HR and CR functional services had a value of more than CNY 2000 billion, accounting for 55.23% of the total value of ecosystem services. The highest value of HR functional services was CNY 3614.90 billion, followed by a CR functional service value of CNY 2571.14 billion. The functional service values of GR, NC, and SC were between CNY 1000 and 2000 billion, and the value of the three individual service functions accounted for 30.01%. The functional values of SC and GR were quite different, at about CNY 317.19 billion. The functional service values of FD, RMP, WT, BP, and PAL did not exceed CNY 1000 billion, of which the higher functional service value of WT was CNY 958.15 billion, and the lowest functional service value of BP was CNY 100.73 billion, accounting for 0.87% of the total ESV. The value of the individual functions of TR and TRFP was high, among which, the sum of the functional values of CR, HR, SC, NC and PAL accounted for more than 50% of the total value of individual functions, and the FD functional value of TR and TRFP accounted for only 35.84% of the total value of FD functions. The value of each individual ecosystem service function of LLM was less than 2%.

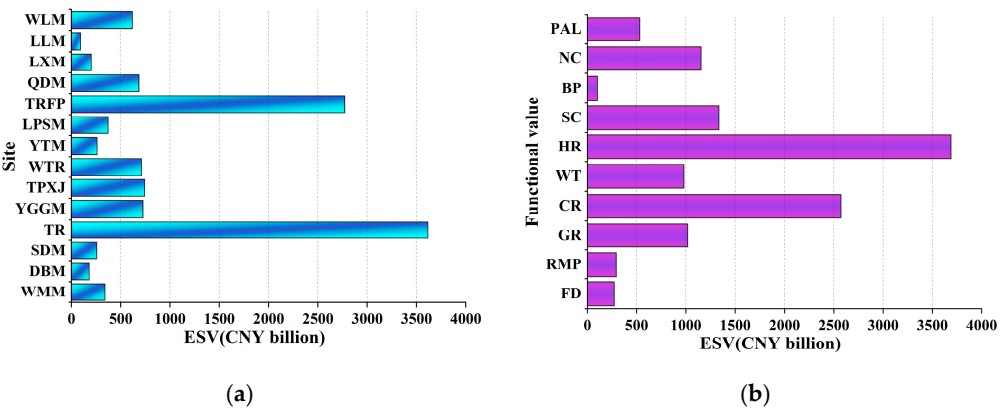

**Figure 3.** Current status of ESV in 2022 ((**a**)The ESV in different areas; (**b**) the functional value of individual ecosystem services).

### 4.3. ESV Characteristics of Different Land Use Types

The ESV varied significantly depending on the land use type. In 2015 and 2022, forest, grassland, and wetland dominated the OCPSAC, together accounting for more than 91% of the total ESV. The ESV of grassland was the highest, more than CNY 4300 billion, accounting for about 38% of the total ESV. The ESV in each area was more than CNY 5 billion. The grassland ecosystem services of TR and TRFP were much higher than that of other areas, accounting for 71.47% of the grassland ESV. The ESV of grassland in DBM was the lowest in the OCPSAC, at about CNY 6.90 billion (Figure 4a). The forest ESV was relatively high and accounted for about 36% of the total ESV. The functional ESV in each area exceeded CNY 10 billion, and the ESVs of YGGM, WYB, TRFP, WLM and TR exceeded CNY 500 billion, accounting for 71% of the forest ESV. The ESV of ores in TPXJ was the lowest in the OCPSAC, at about CNY 12 billion (Figure 4b). The wetland ESV accounted for about 17% of the total ESV in the OCPSAC and the functional ESV in each area exceeded CNY 3 billion. The wetland ESV of TR was more than CNY 1000 billion, and the wetland

ESVs of TRFP and TPXJ were more than CNY 200 billion (Figure 4c). The farmland ESV in each area was more than CNY 5 billion. For example, the farmland ESVs of QDM, LPSM, and SDM were more than CNY 50 billion, and the minimum value of farmland ESV in TR was about CNY 9.64 billion (Figure 4d). The desert ESV was low, and the desert ESVs of QDM, TR, and TPXJ ESV accounted for 97% of the total desert ESV (Figure 4e). After 2022, the ESV of different land use types is still increasing (Figure 4f).

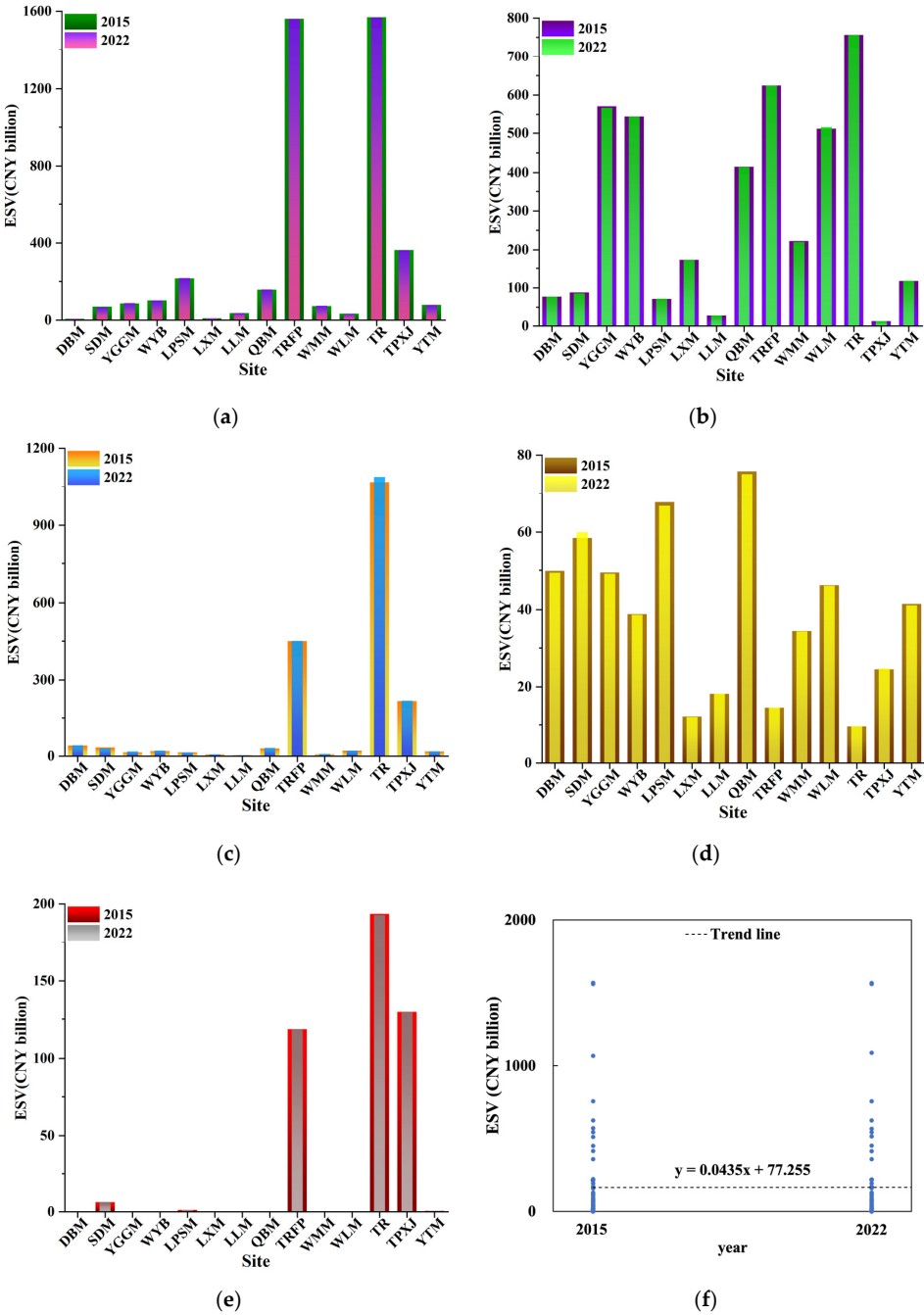

**Figure 4.** The ESVs in the OCPSAC of different land use types ((**a**) grassland; (**b**) forest; (**c**) wetland; (**d**) farmland; (**e**) desert; and (**f**) the changing trend).

## 4.4. The OCPSAC's ESV Spatial Pattern

In 2015 and 2022, the OCPSAC's ESV was relatively consistent in space, showing characteristics of being high in the middle and low at both ends of the study area (Figure 5a,b). The ESVs were higher in YGGM, WYB, WLM, WMM, LXM, southern TRFP, and eastern

TR, and lower in TPXJ, northern TRFP, western TR, eastern SDM, and northern DBM. The ESV of the OCPSAC was highly consistent with the change in the ladder, showing a spatial distribution pattern with high values in the central area and low values eastern and western areas (Figure 6). The area with a higher ESV was distributed in the second altitude gradient of China (between 500 m and 4000 m above sea level), followed by the third altitude gradient in China (less than 500 m above sea level), while the lowest ESV unit area was in the first altitude gradient (more than 4000 m above sea level). This spatial distribution feature was closely related to the natural and social factors of the OCPSAC. The second altitude gradient was mainly mountainous, plateau, and basin, and was distributed in the south–central part of the study area. Due to its special terrain, the vegetation receives sufficient light, heat, and water, and the natural growth of vegetation is better than in other regions. However, the terrain is dangerous, transportation is inconvenient, and socio-economic development is at a low level. Accordingly, the ecosystem is less disturbed by human activities, which leads to a relatively high ESV. The third altitude gradient is dominated by plains and hills, mainly including SDM and DBM. It has a low terrain, and the soil organic matter content is high, which is conducive to farming. Therefore, human activities are more frequent, the amount of land and resources is large, and the economic development is relatively high, but the vegetation coverage is low, resulting in a low ESV per unit area. The first altitude gradient is dominated by the plateau, mainly including the north of TRFP, TR, and TPXJ. TR and TRFP have a mountain climate, with a high altitude and low temperature. The light and heat conditions required for vegetation growth are poor. Some areas are covered with ice and snow, while TPXJ has poor water resources and land use types of mainly grassland and deserts.

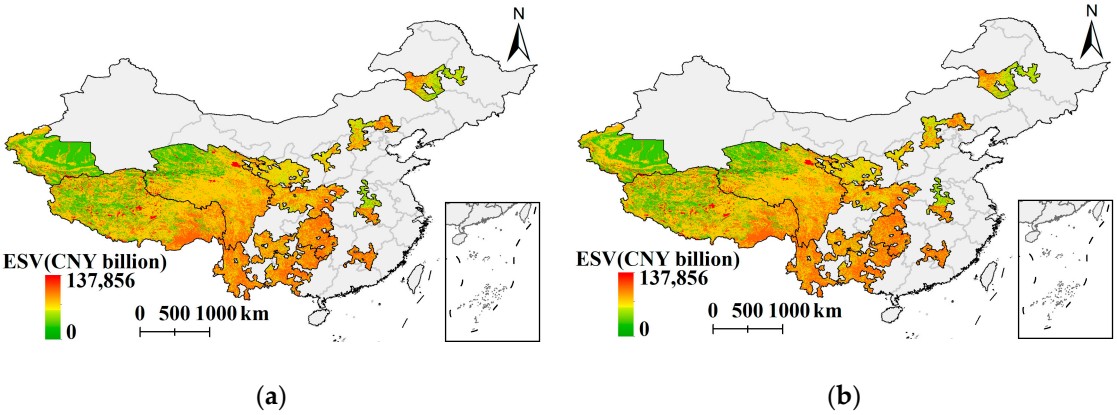

**Figure 5.** Spatial distribution of the ESV in the OCPSAC ((**a**) 2015; (**b**) 2022).

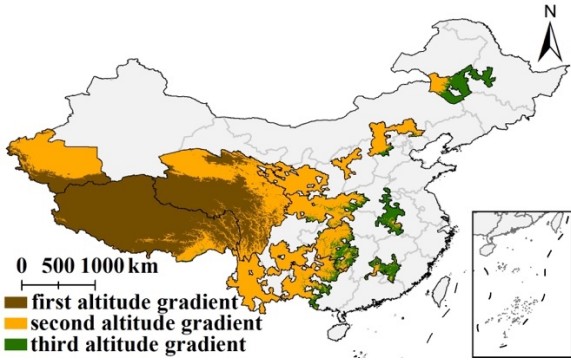

**Figure 6.** Altitude gradient of the OCPSAC.

*4.5. The OCPSAC's ESV Temporal Evolution Characteristics*

In the period of 2015–2022, the OCPSAC's ESV increased, with a total increase of CNY 21.39 billion (Table 3). The ESVs in TR, YGGM, WLM, TPXJ, WYB, YTM, LPSM, TRFP, and QDM showed an increasing trend, among which, TR's ESV increased the most (CNY 17.44 billion). The ESVs in WMM, DBM, SDM, LXM, and LLM were reduced, with the largest reduction in SDM ecosystem services, about CNY 6.32 billion. The functional values of WT, HR, NC, and PAL ecosystem services increased, of which the value of HR functions increased by CNY 21.82 billion, followed by biodiversity and WT functional values, and the minimum increase in the PAL value ecosystem services was about 0.79 billion. The functional values of FD, RMP, GR, CR, SC, and BP decreased, of which the functional value of CR decreased the most, i.e., by about CNY 1.18 billion, and the value of the remaining individual functions was less than CNY 1 billion. In 2022, the ESV of the OCPSAC changed less compared with 2015. The value of each individual function of the LXM decreased and the change in each individual function value was less than 1%. The functional values of LPSM's FD and BP decreased by CNY 0.17 billion, while the functional value of QDM's FD, RMP, GR, and BP decreased by CNY 0.19 billion. In addition, the functional value of TPXJ's CR and SC decreased by CNY 0.12 billion. TR only decreased by 1.16% in the SC function value and WLM decreased by 0.09% in the FD function value. The values of the remaining individual services in LPSM, QDM, TR, WLM, and TPXJ increased. The PAL and HR functional values of DBM and YGGM increased slightly, by CNY 0.06 and 2.12 billion, respectively, and the functional value of LLM and WMM's HR was less than 1%. The FD function value of SDM increased by 1.42%. The sum of the functional values of WT, HR, NC, and PAL 4 of YTM and WYB increased by CNY 0.71 and 2.21 billion, respectively. Finally, the functional value of the remaining individual ecosystem services of DBM, YGGM, LLM, WMM, YTM, SDM, and WYB was reduced. After 2022, the ESV of the OCPSAC showed an increasing trend (Figure 7).

**Table 3.** Functional value of OCPSAC's ESV in 2015 and 2022 (CNY billion).

| Site | Time | FD | RMP | GR | CR | WT | HR | SC | BP | NC | PAL |
|------|------|----|-----|----|----|----|----|----|----|----|-----|
| DBM | 2015 | 11.93 | 5.06 | 17.46 | 30.82 | 10.99 | 64.64 | 15.67 | 2.35 | 13.55 | 6.54 |
|     | 2022 | 11.84 | 5.04 | 17.39 | 30.77 | 10.99 | 64.71 | 15.62 | 2.34 | 13.55 | 6.54 |
| SDM | 2015 | 15.11 | 7.71 | 26.88 | 52.22 | 18.80 | 76.63 | 28.75 | 3.34 | 22.48 | 10.30 |
|     | 2022 | 15.32 | 7.63 | 26.62 | 50.91 | 18.28 | 73.68 | 28.35 | 3.33 | 21.81 | 9.97 |
| YGGM | 2015 | 19.55 | 22.00 | 73.78 | 196.74 | 59.05 | 152.44 | 84.31 | 7.63 | 75.04 | 33.06 |
|      | 2022 | 19.50 | 21.93 | 73.58 | 196.16 | 59.04 | 154.53 | 84.08 | 7.61 | 75.01 | 33.09 |
| WYB | 2015 | 17.35 | 21.18 | 71.10 | 192.47 | 58.18 | 151.30 | 82.18 | 7.22 | 73.94 | 32.62 |
|     | 2022 | 17.32 | 21.17 | 71.06 | 192.39 | 58.27 | 153.25 | 82.13 | 7.21 | 74.04 | 32.69 |
| LPSM | 2015 | 19.42 | 11.57 | 40.66 | 85.43 | 27.96 | 86.55 | 42.56 | 4.73 | 35.74 | 15.72 |
|      | 2022 | 19.26 | 11.59 | 40.71 | 85.93 | 28.12 | 86.92 | 42.69 | 4.72 | 35.96 | 15.82 |
| LXM | 2015 | 5.00 | 6.01 | 20.03 | 54.43 | 16.32 | 43.81 | 23.00 | 2.06 | 20.67 | 9.16 |
|     | 2022 | 4.96 | 5.99 | 19.97 | 54.32 | 16.29 | 43.77 | 22.94 | 2.05 | 20.63 | 9.14 |
| LLM | 2015 | 4.96 | 2.89 | 10.07 | 21.14 | 6.78 | 21.47 | 10.39 | 1.19 | 8.64 | 3.82 |
|     | 2022 | 4.95 | 2.89 | 10.05 | 21.09 | 6.77 | 21.52 | 10.37 | 1.19 | 8.63 | 3.81 |
| QDM | 2015 | 24.56 | 20.65 | 70.25 | 172.57 | 53.47 | 157.88 | 77.13 | 7.65 | 67.99 | 30.16 |
|     | 2022 | 24.44 | 20.63 | 70.21 | 172.67 | 53.58 | 158.96 | 77.15 | 7.64 | 68.12 | 30.24 |
| TRFP | 2015 | 45.22 | 67.54 | 237.47 | 626.34 | 239.48 | 790.19 | 328.79 | 21.85 | 283.43 | 128.35 |
|      | 2022 | 45.24 | 67.56 | 237.54 | 626.52 | 239.62 | 791.24 | 328.94 | 21.86 | 283.57 | 128.43 |
| WMM | 2015 | 11.59 | 10.47 | 35.51 | 89.12 | 27.13 | 72.80 | 39.46 | 3.81 | 34.56 | 15.21 |
|     | 2022 | 11.58 | 10.43 | 35.39 | 88.73 | 27.08 | 73.53 | 39.32 | 3.80 | 34.50 | 15.20 |
| WLM | 2015 | 17.16 | 18.61 | 62.22 | 165.32 | 49.64 | 135.19 | 70.56 | 6.51 | 62.89 | 27.86 |
|     | 2022 | 17.14 | 18.66 | 62.35 | 165.83 | 49.78 | 135.74 | 70.72 | 6.52 | 63.06 | 27.94 |
| TP | 2015 | 51.13 | 75.13 | 265.33 | 693.99 | 299.36 | 1298.77 | 387.69 | 24.22 | 340.78 | 161.04 |
|    | 2022 | 51.24 | 75.15 | 265.40 | 694.16 | 300.12 | 1313.79 | 387.40 | 24.53 | 341.71 | 161.70 |
| TRSX | 2015 | 14.50 | 14.57 | 53.55 | 116.56 | 70.68 | 260.13 | 109.77 | 5.10 | 65.20 | 30.85 |
|      | 2022 | 14.55 | 14.57 | 53.57 | 116.52 | 70.70 | 261.37 | 109.69 | 5.11 | 65.24 | 30.89 |
| TYM | 2015 | 11.93 | 7.89 | 27.18 | 60.88 | 19.49 | 64.02 | 28.76 | 3.13 | 24.68 | 11.01 |
|     | 2022 | 11.86 | 7.87 | 27.12 | 60.86 | 19.52 | 64.66 | 28.72 | 3.12 | 24.71 | 11.03 |

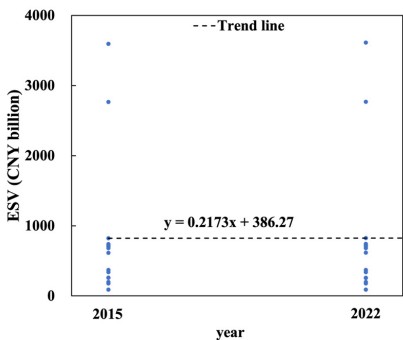

**Figure 7.** Trends in the ESV of the OCPSAC.

*4.6. EC Based on the OCPSAC's ESV*

In 2022, the EC standard of the OCPSAC was CNY 917.14 billion, and the high EC value areas were mainly distributed in TR and TRFP, with compensation amounts of CNY 503.45 and 219.17 billion, respectively, accounting for 72.79% of the total compensation amount. TPXJ, WYB, and YGGM have higher EC values and they accounted for 12.88% of the compensation amount (Figure 8a). The EC amounts of SDM, LPSM, QDM, WMM, and WLM were relatively high, at about CNY 66.92 billion. The EC amounts of DBM, LXM, LLM, and YTM were relatively small, with less than 0.1% of the total EC amount. Among the 14 districts, only the amount of EC in TR exceeded the GDP in 2022, and the amount of EC in the remaining 13 areas was lower than the GDP in 2022, accounting for 4.77% of the total compensation for the special hardship area in 2022 (Figure 8b). In 2015–2022, the OCPSAC's EC standard showed a downward trend, with a decrease of 13.77% (CNY 146.50 billion). The EC amounts in TRFP, WYB, TPXJ, and YGGM were reduced by CNY 108.70 billion, accounting for 74.20% of the total reduction, of which the reduction amount of TREC decreased the most (CNY 62.13 billion). WMM, LPSM, QDM, and TR changed substantially, with a total decrease of CNY 28.96 billion, and other areas changed less. The urgency of these regional EC needs is directly proportional to the ESV in the same period and inversely proportional to the regional GDP in the same period. TR, TRFP, and TPXJ have a high EC priority. This area is in the core area of the Qinghai-Tibet Plateau and represents an ecological barrier. Among them, TR and TRFP are mainly forest land and grassland, each individual ecosystem service function is strong, and the high ESV generated is of great importance for ensuring China's ecological security. Due to the high terrain and inconvenient transportation in TR and TRFP, the TPXJ area is dominated by deserts and slow social and economic development. However, as an important ecological functional area in China, it cannot develop the social economy at the cost of environmental destruction, so it needs financial support to compensate production.

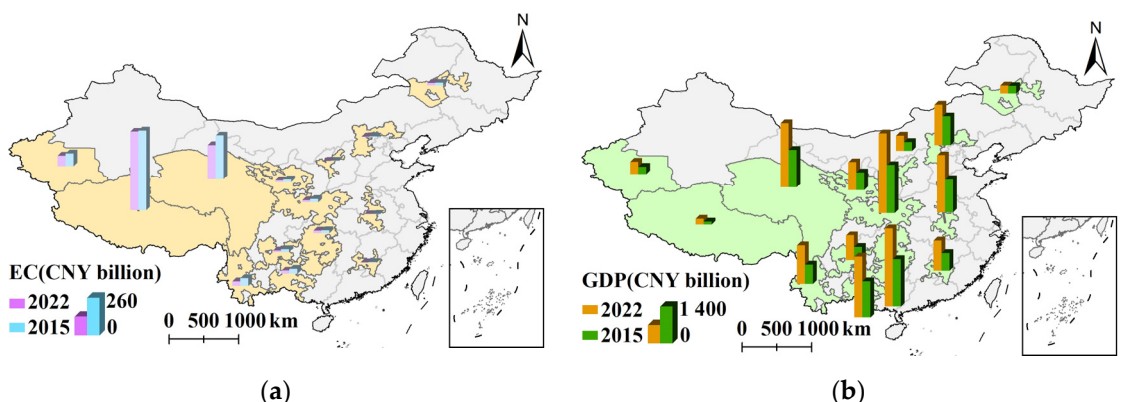

        (**a**)                                     (**b**)

**Figure 8.** Spatial distribution of EC (**a**) and GDP (**b**) in the OCPSAC.

## 5. Discussion

### 5.1. Impact of Land Use Changes on ESV

As an important ecological functional area of the country, the OCPSAC plays an important role in China's homeland security [62–64]. With the development of the social economy, people have conducted large-scale utilization of land resources, leading to higher pressure on the ecological environment [65–67]. This study found that there were subtle changes in the OCPSAC land use between 2015 and 2022, which did not result in significant changes in the ESV. The OCPSAC wetland ESV per unit area was the highest, followed by the forest ESV per unit area. The ESV per unit area was low for other land use categories, which is consistent with the research of scholars such as Bao et al. [68]. Zhang et al. [69] studied the ESV in northwestern Hubei and showed that areas with high ecosystem services per unit area were water, forest, grassland, arable land, and unused land. This is consistent with the results of this study, but due to different land use characteristics in the research area, the division of land use was somewhat different. The change of land use type was positively associated with ESV, which is mainly reflected in the reduction in the areas of farmland, forest, grassland, and desert and the reduction in the ESV. The increase in wetland area led to an increase in the ESV, which is consistent with Wu et al. [70]. Tan et al. [71] showed that increasing the volume of human activities increased the intensity of land use, which significantly reduced the area of grassland and water, and significantly reduced their ESV. The OCPSAC's ESV showed an overall increasing trend, but land use changes in each area were significantly different. For example, TR's ESV increased by 17.44%, while SDM's ESV decreased by 6.32%. Forest, grassland, and wetland areas had higher ESVs, which have different significances for ecological environmental protection.

### 5.2. Impact of ESV on EC

The natural environment and humanistic features of each the OCPSAC section are relatively heterogeneous. Ecosystem services are important providers, which provide various ecological, social, cultural, and economic benefits to society and are the core contents of protection, restoration, and sustainable development [72]. Therefore, we should fully consider the regional uniqueness and differences in the humanistic and natural composition of each area, as well as give a full role to the EC mechanism for promoting and improving the ecological and green development of different regions, which should be further explored. EC is a non-mandatory established standard. Due to the different assessment content and accounting methods, it is difficult to uniformly implement the standard. In the process of implementing EC policies, it is necessary to implement differentiated compensations for different regions. The disbursement of EC funds should prioritize areas with a high ESV but low GDP to prevent an uneven distribution of compensation funds. Zhao et al. [73] also agree with this view. Pan et al. [74] studied the ESV in areas of deep poverty, where Yihan intersects these areas, and showed that it is necessary to understand the regional ecological environment and establish EC mechanisms before carrying out ecological protection work. This contributes to ecological poverty alleviation in regions with backward economic development. Yan et al. [75] believes that the ESV in the upper reaches of the Minjiang River Basin is higher, but its economic development is relatively low, so the EC is also higher. Tian et al. [76] found that EC shows regional differences according to the different spatial distributions of ESV and regional GDP in different years, which is consistent with the results of this study. Guan et al. [77] showed that the ecosystem in China's Yangtze River Basin gradually weakened from east to west in 2015–2017, and TR's EC is low. This is because, from an ecological footprint perspective, the EC quota in areas with intensive human activities should be higher. This study believes that land use layout is one of the influential factors affecting EC. The conversion of its type leads to changes in the ESV and indirectly affects the value of the EC, which is consistent with the results of Yu et al. [78] in the northern section of Taihang Mountain. In the future, we should continue to optimize the spatial allocation of OPCACEC funds and improve the targeting efficiency of regional EC policies.

### 5.3. Sustainable Development of the OCPSAC Based on SDGs

The OCPSAC SDGs are closely linked to several United Nations SDGs [79,80], such as SDG 1 (no poverty) and SDG 2 (zero hunger). Cultivation and specialized fruit industries in the region are the main means of sustaining livelihoods in mountainous areas, and the breeding of local livestock, cultivation of high value-added mountain crops, and sustainable use of natural resources can alleviate food problems [81]. The OCPSAC SDGs are also closely linked to SDG 3 (good health and well-being). The main problems in the OCPSAC are manifested in the prevalence of diseases, stunting, and malnutrition. Therefore, we should intervene in all aspects of health, such as the popularization of health knowledge, rational diet, national fitness, and mental health promotion actions, etc. [82]. In terms of SDG 4 (quality education) and SDG 5 (gender equality), the OCPSAC is geographically poorly located, villages are dispersed, information technology is backward, traditional thinking is strong, and the level of education is relatively low. Therefore, all children should be enabled to complete free, equitable, and quality primary and secondary education and have equal access to vocational and higher education [83]. Regarding SDG 6 (clean water and sanitation), metal ions, pigments, and bacteria in drinking water exceed the standard in some areas. Therefore, water quality should be improved and the integrated management of water resources should be strengthened [84]. The OCPSAC SDGs are closely linked to SDG 8 (decent work and economic growth) and SDG 9 (industry, innovation and infrastructure). The district has poor transportation accessibility, weak and outdated infrastructure, single industry development, and a low level of socio-economic development. Therefore, the infrastructure should be rapidly improved, the ecological environment should be optimized, and the capacity for sustainable development should be significantly enhanced [85]. In terms of SDG 13 (climate action), climate change will lead to a high incidence of extreme weather events in the OCPSAC, which will have a greater impact on agricultural production and livelihoods. Therefore, energy efficiency should be strengthened, clean energy promoted, industrial emissions reduced, vegetation restored, soil improved, and soil and water conserved through natural or artificial means [86]. Finally, regarding SDG 15 (life on land), biodiversity is reduced due to environmental pollution and the invasion of alien species in the study area. This leads to economic losses in agriculture, animal husbandry, and forestry and the weakening of ESV. Therefore, we should protect natural habitats, control species invasion, and reduce ecosystem damage to further maintain ecological balance [87].

### 5.4. The OCPSAC Rural Revitalization Strategy

The OCPSAC implements a rural revitalization strategy, which can guide and support policies and promote the improvement of the agricultural industry, which is not only conducive for transforming its own ecological and resource advantages into industrial and economic advantages, but also provides new economic support points for overall macroeconomic development and promotes the rapid transformation of old and new momentum in economic development [88]. With the development of the economy and society, people have a higher demand for a better life, a stronger desire for the fair enjoyment of excellent ecological products and environments, and increasing expectations for the equalization of public facilities and services. First, the OCPSAC should actively promote rural ecological environmental protection. In the process of promoting the implementation of the rural revitalization strategy, regardless of the ESV level, it is necessary to pay attention to the protection and governance of the rural habitat, constantly innovate and improve the ecological protection and governance mechanism, fully tap into the ecological potential of the OCPSAC, and enable rural revitalization to embark on the path of green and sustainable development [89,90]. Second, the OCPSAC should take full advantage of its rural ecological resources. By analyzing the ESV, this study found that the areas with a higher ESV were TR, TRFP, WYBM, YGGM, QDM, and WLM. "Rich poverty" is the common point of most of the OCPSAC. Most of the OCPSAC has rich ecological resources and distinctive agricultural resources, but cannot be connected to the consumer market due to

geographical barriers and inconvenient transportation. Because capital, talents, technology, and other development elements are scarce, they are unable to be developed. Therefore, the ecological potential of these poverty-stricken areas is enormous, and it is necessary to unleash their ecological potential through open revitalization. The future promotion of rural ecological revitalization should establish the concept of open development, i.e., open to seek breakthroughs and open to achieve development [91]. Finally, the OCPSAC should focus on promoting rural socio-ecological construction, and rural ecological revitalization should address the relationship between economic and ecological development and social and ecological construction [92]. Therefore, more attention should be paid to the often neglected social–ecological construction to maintain the unique form of the countryside, show its charm, and preserve its beautiful ecology.

## 6. Conclusions

The OCPSAC land use types were mainly grassland, forest, and desert, with an area of more than 80% of the study area. In 2015–2022, wetland and other land areas increased, and farmland, forest, grassland, and desert areas decreased. Among them, farmland was mainly transformed into forests and grassland, and other land was mainly transformed into farmland and grassland. The ESV of the OCPSAC reached CNY 11,559.78 billion, and TR and TRFP accounted for 32.28% and 19.64% of the research area, respectively. In a single ESV, HR and CR were the highest and accounted for 55.23% of the research area. In terms of land use types, the ESVs in grassland, forest, and wetland accounted for more than 91% of the research area in 2015 and 2022.

The OCPSAC's ESV was relatively consistent in space, showing characteristics of being high in the middle and low at both ends of the study area. The ESVs of YGGM, WYB, WLM, WMM, and LXM were the highest. In terms of the terrain gradient, the areas with a higher ESV per unit area were distributed on the second altitude gradient, followed by the third altitude gradient, with the lowest ESV per unit area distributed in the first altitude gradient of China. The ESV increased by CNY 21.39 billion in 2015 and 2022. The ESVs in WMM, DBM, SDM, LXM, and LLM decreased, with a total reduction of CNY 7.17 billion. The ESV in the remaining areas increased, with a total increase of CNY 28.46 billion. The functional values of WT, HR, BP, and PAL increased by CNY 24.38 billion, and the functional value of the remaining individual ecosystem services decreased by CNY 3.08 billion.

In 2022, the EC standard of the OCPSAC was CNY 917.14 billion, and the areas with high EC values were mainly distributed in TR and TRFP, accounting for 72.79% of the total compensation amount. The EC amount of TR alone exceeded the GDP in 2022, which accounted for 4.77% of the total compensation of the OCPSAC in 2022. The fragile environment of the OCPSAC is symbiotic with economic poverty. The strategic significance of ecological protection in poverty-stricken areas should be fully considered, and the goal of reducing economic poverty should be achieved by protecting the region's environment.

The assessment of ESV is of great significance for optimizing EC measures, and there is still room for the existing EC policies to continue to be optimized. Since ESV and EC involve many disciplines, and the research content includes social, economic, and ecosystems, it is necessary to improve the accuracy of the assessment of ESV. In addition, there is no unified index system for the evaluation of the ESV. In future ecological research and practice, for the norms, standards, and guidelines for different types of EC, it is necessary to further improve the ESV involved in the EC and their classification standards, and improve the standardization of indicators and evaluation methods.

**Author Contributions:** Data curation, formal analysis, investigation and writing—original draft, Z.Y.; methodology, software, visualization, writing—original draft and writing—review and editing C.S.; supervision, conceptualization and writing—review and editing, H.D. All authors have read and agreed to the published version of the manuscript.

**Funding:** This research was funded by the National Natural Science Foundation of China (NO.42271005).

**Institutional Review Board Statement:** Not applicable.

**Informed Consent Statement:** Informed consent was obtained from all subjects involved in the study.

**Data Availability Statement:** The data presented in this study are available upon request from the corresponding author.

**Conflicts of Interest:** The authors declare no conflicts of interest. The funders had no role in the collection, analysis, or interpretation of the data; in the writing of the manuscript; or in the decision to publish the results.

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
