# Peer review of "Dynamic Changes in Ecosystem Service Value and Ecological Compensation in Original Continuous Poverty-Stricken Areas of China"

_sustainability, doi:10.3390/su16103947_

Round 1
Reviewer 1 Report
Comments and Suggestions for Authors
The authors' article is devoted to an important and topical issue aimed at studying the ecosystem of the region, taking into account socio-economic development.
In scientific research, the essential content of the concept of “ecosystem” varies, depending on the scientific approaches used and the goals of the research. In a regulatory context, the definition of the digital economy ecosystem is presented in the Information Society Development Strategy, according to which “the digital economy ecosystem is a partnership of organizations that ensures the constant interaction of their technological platforms, applied Internet services, analytical systems, and information systems of government authorities. The differentiation of approaches to understanding the essence of ecosystems fully reflects a certain pattern of the dynamic functioning of ecosystems based on a combination of platforms and digital technologies. Based on a number of studies, it is necessary to identify conceptual problems in the formation of systemic ideas about ecosystems of regional development. In modern scientific publications, the development of regional ecosystems is mainly assessed through the prism of assessing the level of digital development of the region. At the same time, an ecosystem as a complex network of interacting organizations based on technologies and platforms is capable of long-term sustainability if the economic and platform interests of the participants in such a network are balanced. It is necessary to introduce the concept of “platform interest”, which is considered, along with economic interest, in order to formulate directions for ensuring the sustainability of ecosystems at the regional level. A conceptual matrix for the development of regional ecosystems should also be presented, reflecting economic and platform interests in the cooperation of stakeholders.
The results obtained in this work will be useful for the analysis of specialists in this field.
1. The introduction could list the main environmental problems occurring in the mountainous regions of China, primarily related to mining waste, the formation of tailings, etc.
2. Figure 1 could be supplemented with a geological map of the area to form a deeper systematic approach to the sustainable development of territories.
3. It should be explained how there were the main five categories when considering land use issues (section “3.1. Data sources and processing”).
4. What are the features of the Google Earth Pro software used, what are its advantages over other similar programs?
5. Based on the data presented in Table 2, it would be necessary to conduct a correlation and regression analysis with the presentation of specific mathematical models that allow for the calculation and prediction of output parameters considered in the context of sustainable development, as well as the corresponding coefficients of determination.
6. Figure 4 should be supplemented with predicted values of the considered output parameters for short-term and long-term periods of time. For forecasting purposes, methods based on the use of artificial intelligence could be used.
7. Based on the research conducted, a generalized research methodology should be presented for which a patent could be obtained.
8. The conclusions should dwell in more detail on the prospects for further research on the presented topics.
Author Response
Thanks to the expert teacher's advice, I have uploaded the modification instructions to the attachment.

Reviewer 2 Report
Comments and Suggestions for Authors
Comments
Based on the improved value coefficient calculation method, the author analyzed the changes in ecosystem service value and ecological compensation in the original contiguous poverty-stricken areas of China. The author comprehensively considered factors such as the level of national economic development and ecological importance in exploring ecological compensation standards and zoning, which is beneficial for enhancing the fairness of ecological compensation policies. The ecological compensation calculation method proposed in this study is somewhat innovative and can serve as a reference for other regions globally. However, there are four issues that need to be addressed, suggesting acceptance upon minor revisions.
(1)In the introduction section, the author mentioned that research on poverty-stricken rural areas in China still relies mainly on quantitative analysis, which contradicts the results presented in numerous existing literature (please refer to the China National Knowledge Infrastructure). Please conduct a new literature review and reconsider the accuracy of the description regarding the research status. Additionally, there is a lack of analysis from the author regarding the current status and issues of research utilizing ecosystem service value for ecological compensation, despite abundant existing studies in this area.
(2)In the data methodology section, there is a lack of explanation regarding the source and accuracy of primary productivity data. Furthermore, it is recommended to provide clarification on the accuracy of remote sensing image interpretation.
(3)The author chose 2015 and 2022 as the basis for calculating ecosystem service value and ecological compensation. Please provide reasons for this choice.
(4)In the discussion section, it is suggested to address potential limitations of this study. For instance, the land use classification is relatively coarse, and different regional single-crop yields were not considered, leading to a need for improved accuracy in evaluating ecosystem service value. Additionally, it is recommended to include future research prospects.
Author Response

(The authors gave the same response as above.)
